# Towards Emotion-enriched Text-to-Motion Generation via LLM-guided Limb-level Emotion Manipulating

Tan Yu
tyu417@stu.suda.edu.cn
School of Computer Science and
Technology, Soochow University
Suzhou, China

Jingjing Wang*
djingwang@suda.edu.cn
School of Computer Science and
Technology, Soochow University
Suzhou, China

Jiawen Wang
20235227102@stu.suda.edu.cn
School of Computer Science and
Technology, Soochow University
Suzhou, China

Jiamin Luo
jmluo97@outlook.com
School of Computer Science and
Technology, Soochow University
Suzhou, China

Guodong Zhou
gdzhou@suda.edu.cn
School of Computer Science and
Technology, Soochow University
Suzhou, China

## Abstract

In the literature, existing studies on text-to-motion generation (TMG) routinely focus on exploring the objective alignment of text and motion, which largely ignore the subjective emotion information, especially the limb-level emotion information. With this in mind, this paper proposes a new **E**motion-enriched **T**ext-to-**M**otion **G**eneration (ETMG) task, aiming to generate motions with the subjective emotion information. Further, this paper believes that injecting emotions into limbs (named intra-limb emotion injection) and ensuring the coordination and coherence of emotional motions after injecting emotion information (named inter-limb emotion disturbance) is rather important and challenging in this ETMG task. To this end, this paper proposes an **LL**M-guided **L**imb-level **E**motion **M**anipulating (L$^3$EM) approach to ETMG. Specifically, this approach designs an LLM-guided intra-limb emotion modeling block to inject emotion into limbs, followed by a graph-structured inter-limb relation modeling block to ensure the coordination and coherence of emotional motions. Particularly, this paper constructs a coarse-grained **Emotional** **T**ext-to-**M**otion (EmotionalT2M) dataset and a fine-grained **Limb**-level **E**motional **T**ext-to-**M**otion (Limb-ET2M) dataset to justify the effectiveness of the proposed L$^3$EM approach. Detailed evaluation demonstrates the significant advantage of our L$^3$EM approach to ETMG over the state-of-the-art baselines. This justifies the importance of the limb-level emotion information for ETMG and the effectiveness of our L$^3$EM approach in coherently manipulating such information.

## CCS Concepts

• **Computing methodologies → Artificial intelligence**.

---

*Corresponding Author: Jingjing Wang.

## Keywords

Emotion-enriched Text-to-Motion, LLM-guided diffusion model, Limb-level emotion manipulating

**ACM Reference Format:**
Tan Yu, Jingjing Wang, Jiawen Wang, Jiamin Luo, and Guodong Zhou. 2024. Towards Emotion-enriched Text-to-Motion Generation via LLM-guided Limb-level Emotion Manipulating . In *Proceedings of the 32nd ACM International Conference on Multimedia (MM '24), October 28-November 1, 2024, Melbourne, VIC, Australia.* ACM, New York, NY, USA, 10 pages. https://doi.org/10.1145/3664647.3681487

## 1 Introduction

Human motion generation leverages various control signals (e.g., music [13, 14], action categories [11, 21] and text descriptions [10, 35, 37]) to generate vivid and realistic motions, which has numerous practical applications in fields such as game production, film and virtual reality. Among these control signals, text is extensively applied as a control condition for motion generation due to its rich semantic details and convenient user interaction, i.e., text-to-motion generation (TMG). Existing studies in TMG focus on using various generative models such as VAEs [10, 22] and diffusion models [5, 30] for the objective alignment of text and motion. However, these studies largely ignore the diverse and subjective emotion information, leading to existing approaches struggling to generate emotion-enriched motions.

Inspired by the above observations, this paper proposes a novel **E**motion-enriched **T**ext-to-**M**otion **G**eneration (ETMG) task, aiming to generate motions with rich emotional expressions, which can significantly enhance audience resonance and emotional connection in the application of many fields, such as virtual avatar [3, 20] and emotional robots [6, 27]. Specifically, the ETMG task generates emotion-enriched motions via given text descriptions. Taking Figure 1 (b) as an example, given the text *"A person, filled with sadness, walks forward."*, the model requires to generate a more realistic and expressive motion sequence with emotion *sad*, compared to the generated motion sequence in Figure 1 (a) with text *"A man walks forward"*. Especially, we explore two kinds of challenges inside the ETMG task.

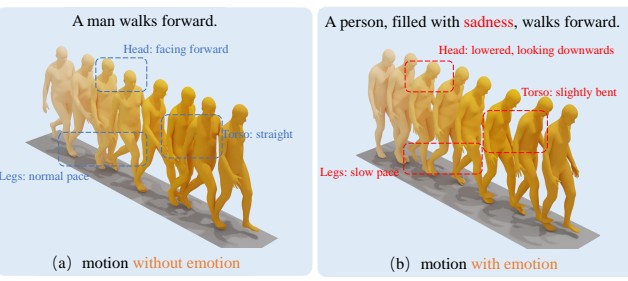

**Figure 1: Two motions generated by our approach. Darker color indicates later frames in the sequence and the red colored word refers to the emotion word. The red and blue box denote limbs have and have not emotional expression.**

For one thing, how to inject emotion into the human limbs is challenging, namely the intra-limb emotion injection challenge. Factually, emotion-enriched motion sequences entail rich emotional semantics, while text description usually contain only one emotion word (e.g., sadness). Also as shown in Figure 1 (b), based on the text description *"A man, filled with sadness, walks forward."*, we obtain the human limbs *Head: lowered, looking downwards; Torso: slightly bent; etc.* with emotion *sad*, where human limbs associated with the motions could correspond to emotional expression in the generated motion sequence. Therefore, a well-designed approach should consider injecting emotion into the human limbs, for precisely matching the human limbs to corresponding emotions and effectively generating emotion-enriched motions.

For another, how to ensure the coordination and coherence of emotional motions after injecting emotion information is challenging, namely the inter-limb emotion disturbance challenge[1]. We assume that the injection of emotion would limit the generation of motions, and affect the coordination and coherence of the generated motions. As shown in Figure 1 (a), the text description *"a man walks forward"* without any emotion generates the motion sequences with no limitations on the human limbs, while after injecting emotion (e.g., *sadness*) into limbs, limbs related to the motion should have corresponding emotional expression, such as *Head: lower; Torso: bent; etc.* Thus, emotion information could cause some disturbance and interference to the motions of the human limbs, which may lead to incoordination and incoherence of the generated emotion-enriched motions. Therefore, a better-designed approach should consider capturing spatial position and relation information between human limbs to tackle emotional disturbance and ensure the coordination and coherence of emotional motions.

In this paper, we propose an **LL**M-guided **L**imb-level **E**motion **M**anipulating (L³EM) approach to tackle the above two challenges. Specifically, this approach first designs an LLM-guided Intra-limb Emotion Modeling (LEM) block via Large Language Models (LLM) to inject emotion into the human limbs to address the intra-limb emotion injection challenge. Then, this approach designs a Graph-structured Inter-limb Relation Modeling (GRM) block via the limb relation graph to capture limb spatial position and relation information for addressing the inter-limb emotion disturbance challenge. Furthermore, a coarse-grained **Emotional T**ext-to-**M**otion

(EmotionalT2M) dataset and a fine-grained **Limb**-level **E**motional **T**ext-to-**M**otion (Limb-ET2M) dataset are constructed to evaluate the effectiveness of the proposed L³EM approach. Detailed experiments demonstrate that our L³EM approach achieves significant improvements compared to the current state-of-the-art baselines.

## 2 Related Work

### 2.1 Text-to-Motion Generation

Text-to-motion generation has always been a popular research topic in generative AI. There is a large body of work proposing different generative neural networks to handle this task. Early studies [1, 9, 22, 29] focus on using joint-latent models to learn a joint embedding space of motion and text. For example, MotionCLIP [29] aligns the human motion manifold to CLIP space implicitly for generating motion sequences with text descriptions. TEMOS [22] introduces a VAE architecture to learn a joint latent space of human motions and text descriptions and can generate different motion sequences given one text description. Inspired by the significant success of diffusion models in the field of image generation [2, 25, 26], some recent works employ diffusion models to handle text-to-motion tasks. For example, MotionDiffuse [36] firstly introduces Denoising Diffusion Probabilistic Models (DDPM) for versatile and controllable human motion generation. MDM [30] predicts the sample, rather than the noise in each diffusion step, thus improving the quality of generated motions. Furthermore, MLD [5] performs a diffusion process on the motion latent space and substantially reduces the computational overhead. Most recently, some works [12, 35, 38] quantize motion sequences into discrete motion tokens and then a GPT-like structure to generate subsequent tokens. For example, MotionGPT [12] uses a Vector Quantised Variational AutoEncoder (VQ-VAE) to quantize motion sequence and use a transformer to automatically generate later tokens.

Although the aforementioned works have made significant progress in the field of text-to-motion, they have always ignored subjective emotion information. Unlike prior studies, this paper first proposes a new Emotion-enriched Text-to-Motion Generation (ETMG) task and proposes two intra-limb emotion injection challenge and inter-limb emotion disturbance challenge in tackling this task.

### 2.2 LLM-enhanced Diffusion Models

Large language models(LLM) have demonstrated impressive capabilities for text generation and semantic understanding [28, 39]. Recently, some works [4, 7, 8, 23, 41] have begun to explore leveraging the powerful abilities of LLM to enhance the generation performance of diffusion models conditioned on text in the fields of Text-to-Image and Text-to-Video. Specifically, in the field of Text-to-Image, Qu et al. [23] use LLM to generate the layout of images and then employ a diffusion model to synthesize high-fidelity images conditioned on both the prompt and the generated layout, which is inspirational to our approach. Zhong et al. [41] propose a Semantic Understanding and Reasoning Adapter (SUR-adapter) for pre-training diffusion models. By transferring knowledge from a large-scale language model (LLM) to the SUR-adapter, the diffusion model is able to understand and reason concise natural language without image quality degradation. In the field of Text-to-Video, Fei et al. [7] take advantage of the existing powerful LLM (e.g.,

---
[1]Comprehensive analysis is described in Section 5.3 "Is emotional disturbance really addressed?"

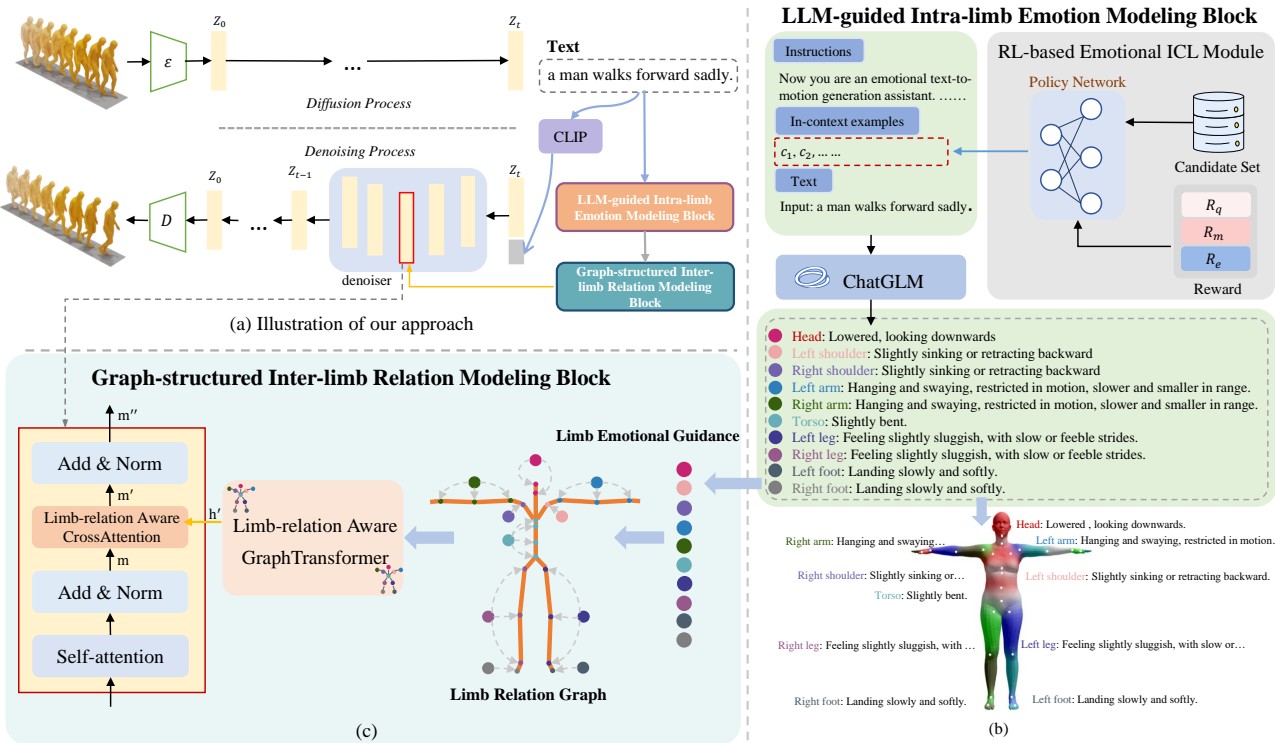

**Figure 2: Our LLM-guided Limb-level Emotion Manipulating (L³EM) approach overview. Wherein (a) is the overall framework of our approach. (b) and (c) is the LLM-guided Intra-limb Emotion Modeling(LEM) Block and the Graph-structured Inter-limb Relation Modeling (GRM) Block, respectively.**

ChatGPT) to generate video dynamic scene graphs for tackling the intricate video temporal dynamics issue in video generation.

Different from all the above studies, this paper first designs an RL-based Emotional ICL module to assist LLM in generating high-quality limb emotional guidance for addressing the intra-limb emotion injection challenge, and then utilizes a limb relation graph to capture limb spatial position and relation information to tackle the emotion disturbance challenge.

## 3 Approach

In this section, we formulate the ETMG task as follows. Given a text description $y$ (e.g., a man walks forward sadly), which describes a motion and conveys an emotion $e$ (e.g., sad), the goal of ETMG is to generate a 3D human motion sequence $\mathbf{x}^{1:L}$ with the length $L$ that matches the text description $y$ and the emotion $e$. In this paper, we propose an L³EM approach for generating motions with rich emotional expressions. The overall framework of L³EM is shown in Figure 2, which mainly comprises the LLM-guided Intra-limb Emotion Modeling (LEM) Block (Sec. 3.2) for addressing the intra-limb emotion injection challenge and the Graph-structured Inter-limb Relation Modeling (GRM) Block (Sec. 3.3) for addressing the inter-limb emotion disturbance challenge.

### 3.1 Preliminary on Latent Diffusion Model

In this paper, we adopt the open-sourced Motion Latent-based Diffusion model (MLD) [5] as our backbone due to its high performance and low computation overhead. MLD first employs a Variational

AutoEncoder (VAE) to obtain a representative and low-dimensional latent code for a motion sequence and then performs a diffusion process on the motion latent space, which substantially improves the computational efficiency. Specifically, a motion sequence $x^{1:L}$ with the length $L$ is first mapped to a motion latent space through motion encoder $\mathcal{E}$ of a pre-trained VAE, i.e., $z_0 = \mathcal{E}(x^{1:L})$. The diffusion process on latent space is modeled as a Markov nosing process using the:

$$q(z_t|z_{t-1}) = \mathcal{N}(\sqrt{\alpha_t}z_{t-1}, (1-\alpha_t)I) \tag{1}$$

where $q(z_t|z_{t-1})$ denotes the posterior distribution. $\alpha_t \in (0, 1)$ is constant hyper-parameters. $\mathcal{N}$ represents the normal distribution and $I$ represents the identity covariance matrix. $\{z_t\}_{t=0}^T$ denotes the noising sequence in the motion latent space, where $T$ represents the time step. Then, at the denoising process, a conditional denoiser $\epsilon_\theta(z_t, t, c)$ is used to predict the noise $\epsilon$ added to $z_{t-1}$ in the diffusion process.i.e.,$\epsilon = \epsilon_\theta(z_t, t, c)$, where $c$ denotes the condition input. In our ETMG task, the condition $c$ is the text embedding encoded by the CLIP [24] text encoder. Subsequently, the denoised version $z_{t-1}$ of $z_t$ at each time step $t$ is obtained by $z_{t-1} = z_t - \epsilon_\theta(z_t, t, c)$. During the inference stage, $z_0$ is obtained with $T$ iterative denoising steps given a noisy latent $z_t \in \mathcal{N}(0, 1)$, timestep $t$, and the conditioning text embedding $c$. Then, motion decoder $D$ of the pre-trained VAE maps $z_0$ to the original motion sequence.

### 3.2 LLM-guided Intra-limb Emotion Modeling

As shown in Figure 2 (b), the LLM-guided Intra-limb Emotion Modeling (LEM) Block is designed to generate limb emotional guidance

from LLM. Specifically, in this paper, we use in-context learning (ICL) [31] to generate emotional guidance for ten major limbs (i.e., head, left shoulder, right shoulder, left arm, right arm, torso, left leg, right leg, left foot, right foot) of humans from LLM (i.e., Chat-GLM [34]). The ICL prompt we use consists of three parts, a task description (Instruction), a few input-output examples (in-context examples), and a test input (Test). Previous studies [18, 19, 40] indicate that the efficiency of ICL is significantly affected by how in-context examples are structured. Therefore, it is crucial to choose appropriate in-context examples from the training set to activate the ICL capability of LLM. In this paper, inspired by [23], we design an RL-based Emotional ICL module in Figure 2 (b) to tackle this issue, where RL and ICL refer to Reinforcement Learning and In-Context Learning, respectively. The RL-based Emotional ICL module mainly consists of two parts, i.e., policy network and emotion-enhanced reward.

• **Policy Network**. Given a text $y$ (e.g., a man walks forward sadly.) and the emotion $e$ (e.g., sad) in the text, we use the policy network $\pi_\theta$ to select K in-context examples $C_I = \{c_k | k = 1, 2, ..., K\}$ from a candidate set $C$, i.e., $c_k \sim \pi_\theta(c|y)$, where we randomly select $N$ samples from training set and annotate them with limb emotional guidance, forming the candidate set $C$. $c_k \in C$ is independently sampled from the candidate set $C$. Specifically, the policy network $\pi_\theta$ is implemented as follows:

$$\pi_\theta(c|y) = \frac{\exp(f(y[c]) \cdot f(y) + f(e[c]) \cdot f(e))}{\sum_{c' \in C} \exp(f(y[c']) \cdot f(y) + f(e[c']) \cdot f(e))} \quad (2)$$

where $y[c]$ and $e[c]$ denote the text (e.g., A man steps forward, looking very happy.) and the emotion (e.g., happy) concerning the candidate $c$. $f(\cdot)$ represents a mapping function that transforms a text and emotion into a latent embedding. In latent space, sentences with similar semantics and close emotions are to be mapped close to each other.

• **Emotion-enhanced Reward**. In this paper, our goal is on the one hand to generate motions that are of high quality and match the text, and on the other hand, we also want to make sure that the generated motions are emotion-enriched. Therefore, we design an Emotion-enhanced reward $R$, which consists of three main parts, detailed as follows:

$$R = \alpha R_q + \beta R_m + \gamma R_e \quad (3)$$

where $\alpha$, $\beta$, $\gamma$ are hyper-parameters. $R_q$, $R_m$, and $R_e$ are used to measure the quality of generated motions, the matching with the text, and how much the generated motions correspond to the intended emotions, respectively. Specifically, they are calculated by:

$$R_q = \text{Cos}(\text{MotionEncoder}(\hat{x}^{1:L}), \text{MotionEncoder}(x^{1:L}))$$

$$R_m = \text{Cos}(\text{MotionEncoder}(\hat{x}^{1:L}), \text{TextEncoder}(y)) \quad (4)$$

$$R_e = \text{WF1} + \text{NN-WF1}$$

where Cos denotes the cosine similarity between two vectors. The larger the value, the closer the distance between the two vectors. MotionEncoder and TextEncoder denote the motion encoder and text encoder in our trained evaluator (Detailed in Sec 4.2), which aims at encoding the paired text and motion sequences into a joint embedding space. $\hat{x}^{1:L}$ and $x^{1:L}$ represent the generated motion and the ground-truth motion, respectively. $y$ denotes the text description corresponding to the motion. WF1 and NN-WF1 (Detailed in Sec 4.2)

is the overall classification weighted average f1-score and the non-neutral classification weighted average f1-score, which reflects how much the generated motions correspond to the intended emotion in all emotion classes and non-neutral emotion classes, respectively.

Through the RL-based Emotional ICL module, we select the K in-context examples $C_I$. Then, we combine it with instruction and $y$, inputting them into ChatGLM [34] to obtain the limb emotional guidance $G = \{g_1, g_2, ..., g_{10}\}$, i.e., $G = \text{ChatCLM}(\text{Instruction}, C_I, y)$, where $g_i, i \in \{1, 2, .., 10\}$ denotes the emotional guidance for each limb (i.e., head, left shoulder, right shoulder, left arm, right arm, torso, left leg, right leg, left foot, right foot).

## 3.3 Graph-structured Inter-limb Relation Modeling

In the above LEM Block, we use LLM (i.e., ChatGLM [34]) to generate emotional guidance for each limb. However, this emotional guidance lacks spatial position and relation information about the limbs, which is crucial for guiding diffusion models in generating co-ordinated and coherent emotional motions. In this paper, as shown in Figure 2 (c), we integrate a limb relation graph into a Graph-Transformer(namely the Limb-relation Aware GraphTransformer) to capture this information.

Specifically, for each limb (i.e., head, left shoulder, right shoulder, left arm, right arm, torso, left leg, right leg, left foot, right foot) emotional guidance $g_i$, where $i \in \{1, 2, .., 10\}$, we first use the CLIP text encoder $C_T$ to convert it into a vector representation $\mathbf{f} = \{C_T(g_1), C_T(g_2), ..., C_T(g_{10})\}$. Then, we use the skeleton structure of SMPL [16] with 22 joints as our limb relation graph. In addition, also as shown in Figure 2 (c), we use linear mapping to project these vector representations $\mathbf{f}$ onto the corresponding joints as the initialization for graph nodes(i.e., $\mathbf{h} = \text{Linear}(\mathbf{f})$, where $\mathbf{h}$ denotes the initial vector representation of the graph nodes. Linear refers to the linear mapping.). Subsequently, the limb relation graph is fed into a GraphTransformer to capture spatial position and relation information about the limbs based on the emotional guidance of limbs.

$$\mathbf{h}' = \text{GraphTransformer}(\mathbf{h}, \mathbf{A}) \quad (5)$$

where GraphTransformer denotes a graph transformer network [33]. $\mathbf{A}$ refers to the adjacency matrix of the limb relation graph. $\mathbf{h}'$ represents the output representation vector of the GraphTransformer, which integrates both spatial position and relation information and emotional guidance information of limbs.

In order to integrate the output representation vector $\mathbf{h}'$ into the transformer layer in denoiser, we introduce a Limb-relation Aware CrossAttention. Specifically, as shown in Figure 2 (c), the output motion feature $\mathbf{m}''$ of the transformer layer is obtained by:

$$\mathbf{m}' = \text{CrossAttn}(\mathbf{m}, \mathbf{h}', \mathbf{h}')$$
$$\mathbf{m}'' = \text{LayerNorm}(\mathbf{m} + \mathbf{m}') \quad (6)$$

where CrossAttn denote the Cross-Attention. LayerNorm denotes the layer normalization. $\mathbf{m}$ denotes the intermediate motion feature.

## 3.4 Optimization for L³EM

To ensure the stability of reinforcement learning training, we employ a two-stage training strategy. Specifically, in the first stage,

**Table 1: Statistics of our Emotional Text-to-Motion ( EmotionalT2M) dataset.**

| Data | #Happy | #Surprise | #Neutral | #Sad | #Angry | #Fear | #Disgust | #Contempt | Total |
|------|--------|-----------|----------|------|--------|-------|----------|-----------|-------|
| Texts | 2300 | 1217 | 4530 | 3386 | 1944 | 367 | 839 | 209 | 14792 |
| Motions | 2199 | 1214 | 4524 | 3349 | 1883 | 336 | 839 | 209 | 14553 |

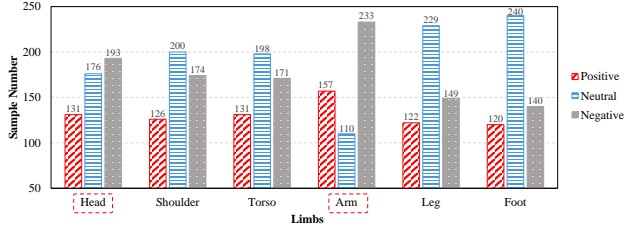

**Figure 3: Statistics of our Limb-level Emotional Text-to-Motion (Limb-ET2M) dataset. The red box represents the two limbs with the least neutral emotion in the dataset.**

we optimize our model by the simple objective used in MLD [5]:

$$\mathcal{L}_{\epsilon \sim \mathcal{N}(0,1), t \in [1,T]} = \mathbb{E}_{\epsilon, t, c} \left[ \| \epsilon - \epsilon_\theta (z_t, t, c) \|_2^2 \right] \tag{7}$$

This optimization objective is designed to enable the denoiser $\epsilon_\theta (z_t, t, c)$ to accurately predict the noise $\epsilon$ added during the diffusion stage.

In the second stage, we use reinforcement learning to further improve the performance of our L³EM model. We optimize the policy network $\pi_\theta$ with policy gradient algorithm [32]. In detail, we first obtain the reward $R$ according to Eq.(3). Then, the policy gradient is computed by differentiating the maximized expected reward $\nabla_\theta J(\theta)$ as follows:

$$\nabla_\theta J(\theta) = \mathbb{E}_{c \sim \pi_\theta(c|y)} \nabla_\theta \log(\pi_\theta(c|y)R) \tag{8}$$

## 4 Experimental Settings

### 4.1 Dataset Construction

To evaluate the effectiveness of our approach to the ETMG task, we construct a coarse-grained **Emo**tional **T**ext-to-**M**otion (EmotionalT2M) dataset and a fine-grained **Limb**-level **E**motional **T**ext-to-**M**otion (Limb-ET2M) dataset.

• **Coarse-grained EmotionalT2M dataset.** The coarse-grained EmotionalT2M dataset is constructed based on the HumanML3D dataset [10] and IDEA400 motion dataset [15]. Specifically, **1) for the Humanml3D dataset**, it consists of 14616 motions and 44970 texts. However, it lacks emotion labels, and only a small portion of the data contains emotion information. Therefore, in order to construct our EmotionalT2M dataset, we manually select data containing emotions and annotate the emotion categories. **2) For the IDEA400 motion dataset**, it comprises 12513 motions and texts. Each motion is labeled with an emotional label, but the text descriptions in the IDEA400 dataset exhibit low quality and lack emotion information. Such as, the text description *Lifting Weights During Walking* lacks a subject and does not convey any emotion. Therefore, in order to construct our high-quality EmotionalT2M dataset, we use ChatGLM [34] to integrate emotion labels(i.e., sad) into the text descriptions, resulting in text descriptions that are grammatically complete and fluent (i.e., A woman is lifting weights while walking, appearing sadly despite her efforts.). Furthermore, in the

IDEA400 motion dataset, motion is represented using tomato representation [17], which is different from the motion representation used in HumanML3D. To merge the two datasets, we convert the tomato representation used in the IDEA400 motion dataset to the motion representation used in HumanML3D. **3) In the end, we obtain our EmotionalT2M dataset**, which comprises 14553 motions and 14792 text descriptions, with 8 emotion categories (i.e., Happy, Surprise, Neutral, Sad, Angry, Fear, Disgust, Contempt). Same as the HumanML3D dataset, we randomly split the dataset into training, validation, and test sets in a ratio of 0.8: 0.15 : 0.05. Detailed statistics of EmotionalT2M dataset are shown in Table 1.

• **Fine-grained Limb-ET2M dataset.** In order to validate that our approach can generate emotion-enriched motions for all limbs (i.e., head, shoulder, torso, arm, leg, and foot), we additionally annotate a fine-grained **Limb-level E**motional **T**ext-to-**M**otion (Limb-ET2M) dataset in the test set of our EmotionalT2M dataset. Specifically, we first spend about a month rendering a total of 500 emotional motions using Blender[2]. Then, we assign two annotators to annotate three emotions for each limb of every motion sequence. Taking the motion sequence shown in Figure 1 (b) as an example, all the limbs in this motion exhibit negative emotion expressions. Therefore, we label all his limbs as *"negative"*. The *Kappa* consistency check value of the annotation is 0.88. When two annotators cannot reach an agreement, an expert will make the final decision, ensuring the quality of data annotations. The Limb-ET2M dataset comprises 500 motions and 500 text descriptions, with each motion corresponding to a single text description. In addition, we randomly split the dataset into training, validation, and test sets in a ratio of 0.7: 0.1: 0.2, and the statistics of the dataset is shown in Figure 3. From Figure 3, we can observe that head and arm have the least neutral emotion. This aligns with our intuition, i.e., people are more prone to expressing emotions using them compared to other limbs[3].

### 4.2 Evaluation Metrics

Evaluation Metrics are summarized in four parts. **1) Motion Quality:** Frechet Inception Distance (FID) reports quality of the generated motion by calculating the distance between features extracted from real and generated motion sequences. It can also to some extent reflect the coordination and coherence of the generated motions. **2) Condition Matching:** R Precision calculates the text and motion Top 1/2/3 matching accuracy, which reflects the similarity between the text description and the generated motion sequence. Multi-modal Distance (MM Dist) measures the average Euclidean distance between the motion feature and its corresponding text description feature. **3) Generation Diversity:** Diversity(DIV) evaluates the variability of the generated motion sequences, while Multi-Modality (MM) measures the average variance of generated motion sequences within the same text description. **4) Emotion Performance:** We use the overall (e.g., positive, neutral, and negative)

---

[2]https://www.blender.org/download/releases/2-93/
[3]Code and dataset are released at https://github.com/aekx/L3EM

Tan Yu, Jingjing Wang, Jiawen Wang, Jiamin Luo, & Guodong Zhou

**Table 2: Results on the EmotionalT2M dataset. All methods use the real motion length from the ground truth for a fair comparison. '↑'('↓') indicates that the values are better if the metric is larger (smaller). We run all the evaluations 20 times. $x^{\pm y}$ indicates that the average metric is $x$ and the the 95% confidence interval is $y$.**

| Approach | R Precision↑ | | | FID↓ | MM Dist↓ | Diversity↑ | MultiModality↑ | WF1↑ | NN-WF1↑ |
|---|---|---|---|---|---|---|---|---|---|
| | Top 1 | Top 2 | Top 3 | | | | | | |
| Real motions | $0.365^{\pm.003}$ | $0.572^{\pm.002}$ | $0.707^{\pm.002}$ | $0.005^{\pm.000}$ | $4.134^{\pm.001}$ | $8.512^{\pm.055}$ | - | $0.509^{\pm.000}$ | $0.523^{\pm.000}$ |
| T2M | $0.214^{\pm.003}$ | $0.326^{\pm.002}$ | $0.432^{\pm.002}$ | $1.065^{\pm.020}$ | $4.857^{\pm.007}$ | $10.293^{\pm.094}$ | $1.324^{\pm.060}$ | $0.371^{\pm.003}$ | $0.387^{\pm.002}$ |
| MotionDiffuse | $0.299^{\pm.003}$ | $0.407^{\pm.003}$ | $0.545^{\pm.003}$ | $0.893^{\pm.021}$ | $4.912^{\pm.012}$ | $9.012^{\pm.015}$ | $2.031^{\pm.069}$ | $0.381^{\pm.001}$ | $0.396^{\pm.002}$ |
| MDM | $0.211^{\pm.005}$ | $0.342^{\pm.006}$ | $0.456^{\pm.006}$ | $1.759^{\pm.034}$ | $6.314^{\pm.040}$ | $8.886^{\pm.013}$ | $4.838^{\pm.188}$ | $0.398^{\pm.005}$ | $0.415^{\pm.004}$ |
| MotionGPT | $0.218^{\pm.002}$ | $0.329^{\pm.005}$ | $0.500^{\pm.003}$ | $0.946^{\pm.013}$ | $6.161^{\pm.027}$ | $7.638^{\pm.053}$ | $1.677^{\pm.087}$ | $0.377^{\pm.004}$ | $0.395^{\pm.002}$ |
| T2M-GPT | $0.235^{\pm.004}$ | $0.385^{\pm.003}$ | $0.537^{\pm.004}$ | $0.800^{\pm.022}$ | $5.974^{\pm.016}$ | $7.948^{\pm.068}$ | $1.801^{\pm.050}$ | $0.389^{\pm.003}$ | $0.411^{\pm.001}$ |
| MLD | $0.248^{\pm.002}$ | $0.403^{\pm.002}$ | $0.559^{\pm.002}$ | $0.768^{\pm.018}$ | $4.840^{\pm.002}$ | $8.365^{\pm.043}$ | $1.639^{\pm.072}$ | $0.403^{\pm.002}$ | $0.419^{\pm.003}$ |
| ReMoDiffuse | $0.281^{\pm.002}$ | $0.447^{\pm.002}$ | $0.567^{\pm.002}$ | $0.117^{\pm.003}$ | $4.833^{\pm.006}$ | $8.440^{\pm.062}$ | $0.750^{\pm.022}$ | $0.417^{\pm.005}$ | $0.428^{\pm.006}$ |
| **L³EM (ours)** | $\mathbf{0.293^{\pm.002}}$ | $\mathbf{0.455^{\pm.003}}$ | $\mathbf{0.607^{\pm.003}}$ | $\mathbf{0.488^{\pm.023}}$ | $\mathbf{4.403^{\pm.003}}$ | $\mathbf{8.679^{\pm.081}}$ | $\mathbf{1.758^{\pm.066}}$ | $\mathbf{0.445^{\pm.002}}$ | $\mathbf{0.463^{\pm.003}}$ |
| w/o LEM | $0.241^{\pm.003}$ | $0.411^{\pm.002}$ | $0.554^{\pm.002}$ | $0.783^{\pm.027}$ | $4.996^{\pm.002}$ | $8.284^{\pm.077}$ | $1.534^{\pm.058}$ | $0.396^{\pm.006}$ | $0.413^{\pm.005}$ |
| w/o RL | $0.278^{\pm.003}$ | $0.445^{\pm.002}$ | $0.589^{\pm.003}$ | $0.594^{\pm.028}$ | $4.537^{\pm.003}$ | $8.475^{\pm.063}$ | $1.702^{\pm.061}$ | $0.432^{\pm.002}$ | $0.454^{\pm.003}$ |
| w/o GRM | $0.265^{\pm.002}$ | $0.437^{\pm.003}$ | $0.566^{\pm.003}$ | $0.612^{\pm.030}$ | $4.815^{\pm.002}$ | $8.377^{\pm.061}$ | $1.664^{\pm.049}$ | $0.417^{\pm.001}$ | $0.435^{\pm.002}$ |

classification weighted average f1-score (WF1) and the non-neutral (e.g., positive and negative) classification weighted average f1-score (NN-WF1) to evaluate the emotional performance of generated motions. Specifically, we first train a classifier based on the real motion sequence and emotion labels in our EmotionalT2M dataset. The trained classifier can accurately classify the emotion labels of motions. Subsequently, we classify our generated motions using the trained classifier, and the resulting WF1 score reflects how much the generated motions correspond to the intended emotions in all emotion classes (i.e., Happy, Surprise, Sad, Angry, Fear, Disgust, Contempt, Neutral), the NN-WF1 score reflects how much the generated motions correspond to the intended emotions in the non-neutral emotion classes (i.e., Happy, Surprise, Sad, Angry, Fear, Disgust, Contempt). Similarly, based on the annotation information in Figure 3, we separately train six emotion classifiers for six limb parts (i.e., head, should, torso, arm, leg, foot) on the Limb-ET2M dataset. Then, we input the generated motion sequences of each limb (e.g., arm motion sequences) into the corresponding emotional classifier (e.g., arm emotion classifier) and obtain the WF1 score and the NN-WF1 score, which reflects how much the generated motions of each limb correspond to the intended emotions in all emotion classes (i.e., positive, neutral and negative) and how much the generated motions of each limb correspond to the intended emotions in the non-neutral emotion classes (i.e., positive and negative).

**For the Motion Quality, Generation Diversity, and Condition Matching**, following Guo et al. [10], we train our contrastive model on our EmotionalT2M dataset as the evaluator. Specifically, this contrastive model consists of a motion encoder and a text encoder, which aims to map the paired text descriptions and motion sequences into a joint embedding space. For the motion encoder, we use the same motion encoder as employed by Guo et al. [10]. For the text encoder, we employ the text encoder used in our approach (i.e., CLIP text encoder). We train the contrastive learning model with the same loss in Guo et al. [10]. **For the Emotion Performance**, previous studies [10, 37] on motion generation don't evaluate the emotional performance of generated motions. For the first time, we use emotion classification metrics (i.e., weighted average f1-score)

to assess the emotional performance of generated motions and train seven emotion classifiers(i.e., the overall motion emotion classifier and the six motion emotion classifier for six limbs) as evaluators.

## 4.3 Baselines

We choose several advanced baselines in Text-to-Motion Generation (TMG) task to compare performance with our approach, described as follows. **T2M** [10] proposes a two-stage approach: text2length sampling and text2motion generation to tackle the TMG task. **MotionDiffuse** [36] is the first diffusion model-based text-driven motion generation framework. **MDM** [30] predicts sample rather than noise in each diffusion step to facilitate the use of geometric losses. **MLD** [5] designs a powerful VAE to get the low-dimensional latent codes for human motion sequences and then performs a diffusion process on the motion latent space. **MotionGPT** [38] can generate consecutive human motions by interpreting multimodal signals as unique input tokens within LLM. **T2M-GPT** [35] uses VQ-VAE and GPT for the TMG task. **ReMoDiffuse** [37] integrates a retrieval mechanism to refine the denoising process and enhances the generalizability and diversity of text-driven motion generation. This model is the state-of-the-art model in text-to-motion tasks.

## 4.4 Implementation Detail

In our experiments, we re-implement all the baselines on our ETMG datasets according to their open-source codes. Besides, the hyperparameters of these baselines reported by their public papers are still adopting the same setting. The others and the hyper-parameters of our L³EM approach are tuned according to the validation set. In the first stage, we employ a frozen CLIP-ViT-L-14[4] model as the text encoder. Adam is adapted as the optimizer to train the model with a learning rate equal to 6e-5. The batch size and the epoch are 32 and 5k, respectively. We use ChatGLM (130B[5]) [34] to generate limb emotional guidance. In the second stage, The $\alpha$, $\beta$ and $\gamma$ in Eq.(3) is 0.5, 0.5 and 0.7, respectively. The size of candidate $C$ is 32 (i.e., $N = 32$), and we set 3 as the shot number (i.e., $K = 3$)

---

[4]https://huggingface.co/openai/clip-vit-large-patch14
[5]https://www.zhipuai.cn/

**Table 3: Results on our Limb-ET2M Dataset to evaluate the emotional performance of each limb.**

| Approach | Head | | Shoulder | | Torso | | Arm | | Leg | | Foot | |
|---|---|---|---|---|---|---|---|---|---|---|---|---|
| | WF1 | NN-WF1 | WF1 | NN-WF1 | WF1 | NN-WF1 | WF1 | NN-WF1 | WF1 | NN-WF1 | WF1 | NN-WF1 |
| Real motions | $0.727^{\pm.000}$ | $0.786^{\pm.000}$ | $0.712^{\pm.000}$ | $0.773^{\pm.000}$ | $0.721^{\pm.000}$ | $0.775^{\pm.000}$ | $0.731^{\pm.000}$ | $0.789^{\pm.000}$ | $0.714^{\pm.000}$ | $0.772^{\pm.000}$ | $0.709^{\pm.000}$ | $0.761^{\pm.000}$ |
| T2M | $0.630^{\pm.002}$ | $0.688^{\pm.003}$ | $0.601^{\pm.003}$ | $0.647^{\pm.004}$ | $0.573^{\pm.001}$ | $0.635^{\pm.003}$ | $0.603^{\pm.002}$ | $0.661^{\pm.002}$ | $0.614^{\pm.003}$ | $0.653^{\pm.002}$ | $0.598^{\pm.003}$ | $0.644^{\pm.005}$ |
| MotionDiffuse | $0.655^{\pm.003}$ | $0.700^{\pm.002}$ | $0.642^{\pm.003}$ | $0.685^{\pm.003}$ | $0.609^{\pm.005}$ | $0.669^{\pm.003}$ | $0.627^{\pm.002}$ | $0.696^{\pm.006}$ | $0.648^{\pm.001}$ | $0.692^{\pm.004}$ | $0.611^{\pm.002}$ | $0.669^{\pm.003}$ |
| MDM | $0.641^{\pm.003}$ | $0.693^{\pm.004}$ | $0.630^{\pm.002}$ | $0.679^{\pm.003}$ | $0.621^{\pm.004}$ | $0.683^{\pm.002}$ | $0.643^{\pm.003}$ | $0.685^{\pm.005}$ | $0.647^{\pm.002}$ | $0.684^{\pm.003}$ | $0.627^{\pm.001}$ | $0.662^{\pm.006}$ |
| MotionGPT | $0.631^{\pm.002}$ | $0.682^{\pm.004}$ | $0.622^{\pm.001}$ | $0.673^{\pm.002}$ | $0.613^{\pm.003}$ | $0.675^{\pm.004}$ | $0.629^{\pm.005}$ | $0.676^{\pm.004}$ | $0.637^{\pm.006}$ | $0.680^{\pm.005}$ | $0.629^{\pm.003}$ | $0.667^{\pm.007}$ |
| T2M-GPT | $0.639^{\pm.004}$ | $0.699^{\pm.002}$ | $0.635^{\pm.002}$ | $0.682^{\pm.003}$ | $0.625^{\pm.002}$ | $0.686^{\pm.002}$ | $0.645^{\pm.003}$ | $0.692^{\pm.001}$ | $0.641^{\pm.002}$ | $0.691^{\pm.004}$ | $0.634^{\pm.002}$ | $0.683^{\pm.003}$ |
| MLD | $0.658^{\pm.002}$ | $0.711^{\pm.005}$ | $0.651^{\pm.003}$ | $0.699^{\pm.002}$ | $0.655^{\pm.003}$ | $0.706^{\pm.003}$ | $0.658^{\pm.004}$ | $0.714^{\pm.006}$ | $0.652^{\pm.003}$ | $0.696^{\pm.003}$ | $0.656^{\pm.002}$ | $0.698^{\pm.002}$ |
| ReMoDiffuse | $0.674^{\pm.003}$ | $0.733^{\pm.001}$ | $0.666^{\pm.002}$ | $0.721^{\pm.003}$ | $0.672^{\pm.002}$ | $0.725^{\pm.002}$ | $0.673^{\pm.001}$ | $0.731^{\pm.004}$ | $0.665^{\pm.004}$ | $0.719^{\pm.005}$ | $0.671^{\pm.003}$ | $0.720^{\pm.002}$ |
| $L^3$EM (ours) | $\mathbf{0.700^{\pm.003}}$ | $\mathbf{0.757^{\pm.002}}$ | $\mathbf{0.691^{\pm.003}}$ | $\mathbf{0.744^{\pm.003}}$ | $\mathbf{0.696^{\pm.002}}$ | $\mathbf{0.751^{\pm.004}}$ | $\mathbf{0.708^{\pm.003}}$ | $\mathbf{0.762^{\pm.003}}$ | $\mathbf{0.692^{\pm.001}}$ | $\mathbf{0.740^{\pm.002}}$ | $\mathbf{0.692^{\pm.003}}$ | $\mathbf{0.743^{\pm.002}}$ |
| w/o LEM | $0.656^{\pm.002}$ | $0.706^{\pm.004}$ | $0.645^{\pm.004}$ | $0.701^{\pm.003}$ | $0.657^{\pm.001}$ | $0.703^{\pm.002}$ | $0.655^{\pm.003}$ | $0.708^{\pm.005}$ | $0.658^{\pm.003}$ | $0.698^{\pm.004}$ | $0.649^{\pm.005}$ | $0.691^{\pm.003}$ |
| w/o RL | $0.688^{\pm.003}$ | $0.744^{\pm.002}$ | $0.682^{\pm.002}$ | $0.733^{\pm.001}$ | $0.687^{\pm.003}$ | $0.739^{\pm.003}$ | $0.693^{\pm.003}$ | $0.749^{\pm.004}$ | $0.684^{\pm.002}$ | $0.729^{\pm.002}$ | $0.679^{\pm.004}$ | $0.731^{\pm.005}$ |
| w/o GRM | $0.667^{\pm.004}$ | $0.722^{\pm.005}$ | $0.663^{\pm.003}$ | $0.719^{\pm.004}$ | $0.668^{\pm.003}$ | $0.715^{\pm.002}$ | $0.670^{\pm.002}$ | $0.724^{\pm.003}$ | $0.667^{\pm.002}$ | $0.712^{\pm.006}$ | $0.660^{\pm.001}$ | $0.706^{\pm.003}$ |

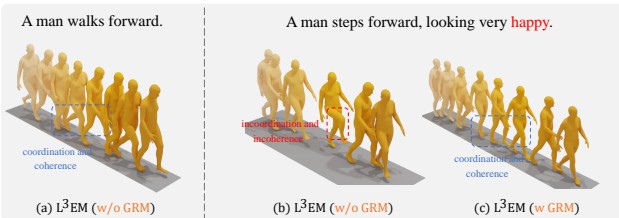

(a) $L^3$EM (w/o GRM)     (b) $L^3$EM (w/o GRM)     (c) $L^3$EM (w GRM)

**Figure 4: Three motions generated by our $L^3$EM approach to illustrate the inter-limb emotion disturbance challenge. The red box indicates the instance of incoordination and incoherence, and the blue box indicates the instance of coordination and coherence.**

by default. In addition, we find that only using few-shot learning can achieve a high performance during this stage. Therefore, we randomly sample 128 instances from the training set for training. The total epochs, batch size, and learning rate are set to 50, 8, and 3e-5, respectively. All experiments are conducted on 1 A100 GPU with 40GB GPU memory.

## 5 Discussion and Analysis

### 5.1 Experimental Results

Table 2 shows the performance comparison of different approaches in the EmotionalT2M dataset. From this table, we can see that: **1) Motion quality.** The emotional motions generated by our approach achieve comparable results in terms of FID compared to other baselines, indicating that the emotional motions generated by our appproach are natural, coherent, and closely resemble real motions. **2) Condition matching.** Our approach outperforms all baselines in R-Precision Top 1 2 3 and MM Dist (e.g., outperforming the state-of-the-art method ReMoDiffuse by 1.2%, 0.8%, 4%, and 0.43%, respectively), indicating that the emotional motions generated by our approach can better match text descriptions, demonstrating superior conditional consistency. **3) Generation diversity.** The Diversity and Multimodality are 8.679 and 1.758, respectively, which are not particularly outstanding compared to baseline methods. We believe this could be attributed to the introduction of emotion information. After introducing emotion information, each limb associated with the motions needs to exhibit a corresponding emotional expression. This, to a certain extent, limits the diversity of generated motions. And since our $L^3$EM approach generates motions with better emotional performance, it consequently results in

lower diversity compared to some baselines. **4) Emotion Performance.** Our approach achieves WF1 score of 44.5% and NN-WF1 of 46.3%, surpassing all current baselines. More importantly, compared to the state-of-the-art baseline ReMoDiffuse, our $L^3$EM approach improves WF1 and NN-WF1 by 2.8% and 3.5%, respectively. This indicates that our approach, integrating limb emotional guidance from LLM and limb relation graph, can generate emotion-enriched motions compared to other baselines.

### 5.2 Are emotions really injected into limbs?

To validate our $L^3$EM approach's ability to inject emotion into human limbs (i.e., the intra-limb emotion injection challenge), we conduct experiments on our Limb-ET2M dataset. Table 3 shows the comparative results. From Table 3, we can see that: **1) WF1:** Our $L^3$EM approach surpasses all the baselines in WF1 score, indicating that our approach could enable all limb to have corresponding emotional expressions compared to the baselines. This justifies our $L^3$EM approach could inject emotions into limbs. **2) NN-WF1:** Our $L^3$EM approach also achieve the SOTA performance in NN-WF1 score. Especially, compared to ReMoDiffuse, our $L^3$EM approach improves the NN-WF1 score of head, shoulder, body, arm, leg, and foot by 2.4%, 2.3%, 2.6%, 3.1%, 2.1%, and 2.3%, respectively. This indicates that, aside from neutral cases, our approach indeed enables limb to exhibit corresponding positive or negative emotional expressions. This further justifies that our $L^3$EM approach could inject emotions into limbs.

### 5.3 Is emotional disturbance really addressed?

To validate our approach's ability to mitigate emotional disturbances and produce coordinate and coherent emotional motions (i.e., the inter-limb emotion disturbance challenge), we conduct qualitative and quantitative experiments. **1) The qualitative results** are shown in Table 2. From this table, we can see that our $L^3$EM approach achieves a comparable FID of 0.488, indicating that the motions generated by our approach are close to real motions, coordinated and coherent. This justifies that our approach addresses the challenge of emotion disturbance. **2) The quantitative analysis** is shown in Figure 4, from Figure 4 (b), we can observe that compared to (a), after introducing emotion information, the generated motion sequence exhibits instances of limb incoordination and incoherence (such as the sudden high leg raise highlighted in the red box in Figure 4 (b)). This validates our argument that emotion information

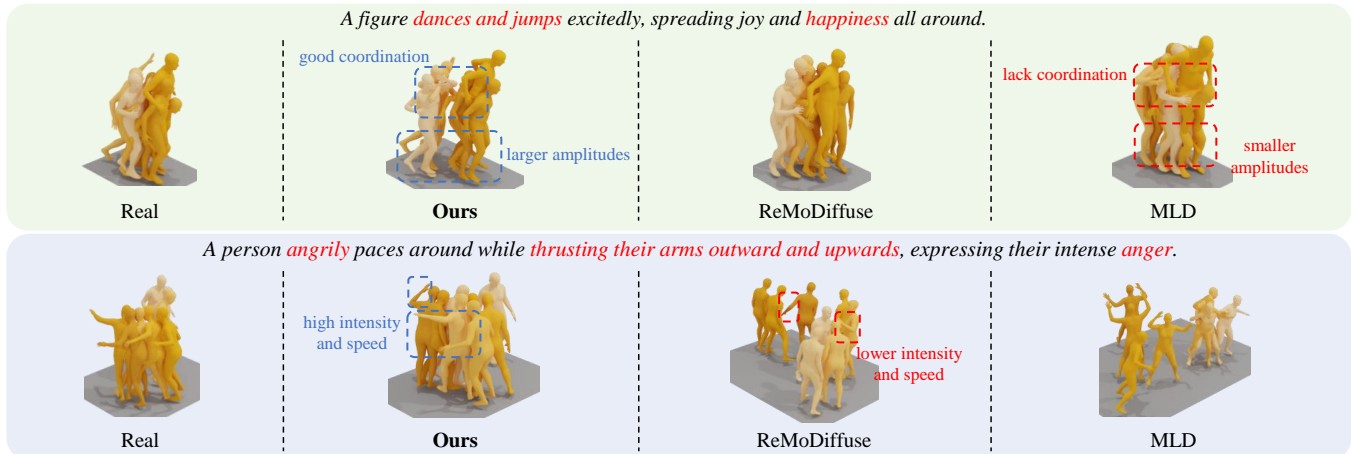

**Figure 5: Qualitative comparison of our** $L^3$EM **approach and the state-of-the-art methods. The blue box and red box reflect the superiority of our** $L^3$EM **approach compared to others.**

can disturb motion generation. Furthermore, from Figure 4 (c), we can observe that after incorporating the limb relationship graph in GRM, compared to Figure 4 (b), our approach is capable of generating coordinated and coherent emotional motions. This further justifies that our approach addresses the challenge of emotional disturbance.

## 5.4 Contributions of Key Components

To further investigate the influence of key components within our $L^3$EM approach, we conduct a series of ablation studies as shown in Table 2 and Table 3. From the tables, we can see that: **1) w/o LEM** exhibits inferior performance compared to $L^3$EM with an increase in FID, and a decrease in R-Precsion Top 1 2 3, WF1, and NN-WF1. This further justifies the effectiveness of integrating limb emotional guidance from LLM in aiding emotional motion generation. Furthermore, the WF1 and NN-WF1 of six limb parts also show inferior performance compared to $L^3$EM. This also justifies that limb emotional guidance indeed facilitates the generation of emotional motions for various limbs, ensuring that each limb of generated motions has corresponding emotional expressions. **2) w/o RL** also display inferior results compared to $L^3$EM in both Table 2 and Table 3. This justifies that our designed RL-based Emotion Modeling module can assist in selecting in-context examples for LLM, thus enhancing the quality of LLM-generated limb emotional guidance and consequently improving the model's generation performance. **3) w/o GRM** yields an inferior performance compared to $L^3$EM with a increase in FID by 0.124. This demonstrates that our proposed limb relation graph can effectively capture spatial position and relation information among limbs, thereby aiding in generating more coordinated and coherent motion sequences.

## 5.5 Qualitative Studies

To further justify the effectiveness of our approach for the ETMG task, we provide a visualization and qualitative analysis as shown in Figure 5. From Figure 5, we can see that MLD [5] and RemoDiffuse [37] are both able to understand the motion description (e.g.,

dance and jump in the example 1) in the text and generate corresponding motions. However, they struggle to have a deep understanding of the emotion information (e.g., spreading joy and happiness all around) within the text. For example, in the first example, the motions generated by MLD exhibit smaller movement amplitudes and lack coordination in limbs, resulting in a weaker expression of happy emotion. In the second example, the motions generated by Remodiffuse, with a lower intensity and speed of arm movement outward and upward, fail to convey the anger emotion effectively. In contrast, our approach, which leverages limb emotional guidance from LLM and limb relation graph, generates emotional motions that not only consist with the text descriptions but also exhibit rich emotional expressions.

## 6 Conclusion

In this paper, we propose an LLM-guided Limb-level Emotion manipulating ($L^3$EM) approach to handle the ETMG task. Additionally, the LEM Block and GRM Block are designed to address the intra-limb emotion injection challenge and the inter-limb emotion disturbance challenges in the ETMG task, respectively. To comprehensively evaluate $L^3$EM, we construct a coarse-grained Emotional Text-to-Motion (EmotionalT2M) dataset and a fine-grained Limb-level Emotional Text-to-Motion (Limb-ET2M) dataset. Experimental results on these datasets demonstrate the superior performance of $L^3$EM over several state-of-the-art baselines. In our future work, we would like to incorporate emotion information into more generative tasks, such as text-to-image and text-to-video tasks. In these tasks, subjective emotion information is also important and has still been underachieving. In addition, we would like to integrate LLM and diffusion more closely, in order to endow our approach with LLM's interactive capability for better emotion manipulating.

## Acknowledgments

We thank our anonymous reviewers for their helpful comments. This work was supported by three NSFC grants, i.e., No.62006166, No.62376178 and No.62076175. This work was also supported by a Project Funded by the Priority Academic Program Development of Jiangsu Higher Education Institutions (PAPD).

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
