# OpenReview forum: "Towards Emotion-enriched Text-to-Motion Generation via LLM-guided Limb-level Emotion Manipulating"
_acmmm.org/ACMMM/2024/Conference — MM2024 Oral_

### Official Review · Reviewer_mtGH · 2024-05-19

**Rating:** 5
**Confidence:** 3

**Summary:**

This paper proposes a new Emotion-enriched Text-to-Motion Generation (ETMG) task, aiming to generate motions with the subjective emotion information. First, few works consider subjective emotion information, especially limb-level emotion information.
Second, injecting emotions into limbs can ensure the coordination and coherence of emotional motions after injecting emotional information.

The motivation is interesting to explore.

**Strengths:**

1. It is an interesting and valuable topic for Text-to-Motion Generation.
2. The writing is good and the authors present the work well.
3. This topic has significant applications in the real world.
4. The authors conducted an extensive experiments.

**Limitations:**

1. The interpretation of experimental analysis needs to be further improved.

**Suitability:**

2

---

### Official Review · Reviewer_TQB5 · 2024-05-23

**Rating:** 6
**Confidence:** 3

**Summary:**

This paper focuses on text-to-motion generation. Their contribution includes incorporating emotion into this task, curating a dataset, LLM-guided emotion manipulating. The task is overall innovative to my knowledge. Incorporating emotion is an interesting direction for text-to-motion generation.

**Strengths:**

(1) This paper introduces a novel task and has dataset contribution.
(2) The methods of LLM-guided intra-limb emotion modeling and graph-structured inter-limb relation modeling are technically reasonable with certain novelty.
(3) The authors performed extensive experiments to show the superiority of their method of incorporating emotion for text-to-motion generation.

**Limitations:**

There are no obvious limitations in this paper.

**Suitability:**

3

---

### Official Review · Reviewer_snck · 2024-05-29

**Rating:** 5
**Confidence:** 2

**Summary:**

This paper first proposes injecting emotion as a condition for motion generation. The authors formulate the emotion-enriched text-to-motion generation task and construct the corresponding dataset. Additionally, they propose a LLM-guided Intra-limb Emotion Modeling Block and a Graph-structured Inter-limb Relation Modeling Block to better embed emotion into the motion generation process. The experimental results demonstrate the effectiveness of the proposed methods.

**Strengths:**

1. The idea of considering emotion in the text-to-motion generation process is interesting.
2. The methods effectively utilize LLM to guide limb generation and Graph-structured Inter-limb Relation Modeling.
3. The experiments and demos show the superiority of the proposed model.

**Limitations:**

1. The dataset is newly constructed. There should be more details regarding the dataset construction process. However, the authors describe the details on pages 5 and 6 with large paragraphs, making comprehension difficult.

2. A question: Does emotion really influence the motion generation results? It is true that different emotional descriptions cause different motions or limb actions. However, the emotion factor is similar to other factors such as walk/run, forward/backward, slowly/quickly, which all influence the final action. The only difference is that the prompt of ChatGLM explicitly emphasizes the emotion factor: "Now you are an **emotional** text-to-motion generation assistant...". If "emotional" is removed from the prompt, does the generation still work? Or if emotion is replaced with another aspect, such as "Now you are a speed-sensitive text-to-motion generation assistant...", will the results also be enhanced?

**Suitability:**

3

---

### Meta-Review · Area_Chair_hL79 · 2024-06-28

**Recommendation:** Accept (Oral)
**Confidence:** 5

**Metareview:**

The paper introduces an innovative approach to text-to-motion generation by incorporating emotion as a condition for generating motions. This novel task, termed Emotion-enriched Text-to-Motion Generation, is supported by a newly constructed dataset. The authors propose two key components: a LLM-guided Intra-limb Emotion Modeling Block and a Graph-structured Inter-limb Relation Modeling Block, which enhance the embedding of emotion into the motion generation process. Reviewers praised the originality and significance of considering emotion in this domain, noting the effectiveness demonstrated through extensive experiments and demos. While the dataset construction process could be more clearly described, and further experimental interpretation is needed, the overall contribution to the field is substantial. Given the strengths and the innovative nature of this work, I recommend acceptance.